# Research on Composite Control Strategy of Quasi-Z-Source DC–DC Converter for Fuel Cell Vehicles

**Meilan Zhou, Mingliang Yang, Xiaogang Wu ***  **and Jun Fu**

School of Electrical and Electronic Engineering, Harbin University of Science and Technology,
Harbin 150080, China
* Correspondence: xgwu@hrbust.edu.cn; Tel.: +86-0451-86391699

**Abstract:** The DC–DC converter for fuel cell vehicles requires high gain and wide voltage input range to boost the voltage of the fuel cell. However, with the traditional boost converter, it is difficult to meet the requirements of the fuel cell vehicle power system. Based on a quasi-Z-source network DC–DC converter, this paper proposes a composite controller, which includes a feedforward compensation network and feedback control to meet the control robustness requirement of the fuel cell vehicle power system. The dynamic model of the converter is obtained by using the state space averaging method and the small-signal dynamic modeling method. The input voltage and load disturbance experiments are performed on the DC–DC converter. Moreover, the converter is tested under the worldwide harmonised light vehicle test procedure (WLTP) to validate the effectiveness of the proposed composite controller. The simulation and experiment results show that the proposed composite controller effectively enhances the converter's ability to resist input and load disturbance, and improves the dynamic response performance of the DC–DC converter for fuel cell vehicles.

**Keywords:** fuel cell vehicles; quasi-Z-source DC–DC converter; feedforward compensation; small-signal dynamic modeling; feedback control

---

## 1. Introduction

The development of the new energy automobile industry brings hope for solving the increasingly prominent environmental problems. Fuel cell vehicles have become a promising research direction in the new energy vehicle industry due to their clean, reliable and efficient advantages [1–3]. However, fuel cells also have some defects. The fuel cell output characteristic is "soft"—the output voltage drops too fast, as the output current increases. The dynamic response speed is slow and cannot follow the rapid variation of electric vehicle demand power. The fuel cell output voltage is low and cannot directly drive the car motor, which requires a high DC bus voltage (e.g., 400 V) [4]. Therefore, in order to provide a stable output voltage to the load, the DC–DC converter with high gain, wide voltage input range and fast dynamic response speed should be used in fuel cell vehicles [5–8].

The isolated DC–DC converter can achieve high gain by changing the turns ratio of the transformer coil. However, due to the leakage inductance of the transformer, a spike voltage absorbing circuit is required and it is also easy to generate additional loss and electromagnetic interference problems [9,10]. Considering the size and cost requirements, the non-isolated DC–DC converter is more suitable for fuel cell vehicles [11]. While the traditional boost converter is simple, low-gain and high voltage stress are not conducive to the components selection. High gain can be achieved by using the cascaded boost converter, but two or more cascaded boost circuits can cause problems such as circuit complexity, high cost and low efficiency [12,13]. The non-isolated step-up DC–DC converters with

switched inductor and switched capacitor are proposed in [14,15] and can achieve high gain. However, the voltage stresses across components are still high. The non-isolated converters proposed in [16,17] can achieve high gain and low voltage stress. However, there is a diode voltage drop between the input and output due to the no-common-ground topology structure. The high frequency pulsating voltage appears between the input and output and easily causes safety problems and electromagnetic interference problems. The converter proposed in [18] has low voltage stress, but the topology has a flying capacitor, which increases the difficulty of control. The multi-level converters are proposed in [19–21] to achieve high gain. However, the complex structure, large size and multiple-switch drive circuits limit the application in fuel cell vehicles. In the research on DC–DC converter feedforward compensation, the traditional boost and buck circuits with the feedforward compensation strategy are proposed in [22–24] and good control effects are obtained. However, these topologies are simple, which are not suitable for fuel cell vehicles.

In [25], a quasi-Z-source DC–DC converter, which has wide voltage input range, high gain, and low voltage stress, is proposed. The theoretical boost ratio can reach 2/(1-2D), where D is the duty cycle. While achieving high boost ratio, the duty cycle is always less than 0.5. However, the disturbance resisting ability and dynamic response performance of a DC–DC converter were not deeply explored in [25]. In the process of fuel cell vehicle driving, the output voltage of the fuel cell always fluctuates with the demand power variation. The output voltage decreases obviously with the increase of output current. The "soft" output characteristic of the fuel cell presents a great challenge to the input disturbance resisting ability of the DC–DC converter. Therefore, it is necessary to improve the input disturbance resisting ability and dynamic response performance of the DC–DC converter for fuel cell vehicles.

The composite controller combining feedforward compensation and feedback control is applied to the quasi-Z-source network DC–DC converter proposed in [25]. The simulation model and 400 W experiment prototype are built to validate the controller by considering the input and output disturbances. The worldwide harmonised light vehicle test procedure (WLTP) is also carried out for DC–DC converters. This paper is organized as follows: In Section 2, the operation principle of the topology is analyzed. The state space averaging method and small-signal dynamic modeling method are adopted to model the converter. In Section 3, the converter parameters are designed and the controller with a feedforward compensation network and feedback control is designed. In Section 4, simulations and experiments are carried out to validate the proposed control strategy. Some conclusions are given in Section 5.

## 2. Topology Operation Principle Analysis and Dynamic Modeling

The DC–DC converter topology based on the quasi-Z-source network is shown in Figure 1. $u_{in}$ is the output voltage of the fuel cell. $D_1$ is the reversed blocking diode, which prevents the current from flowing back to the fuel cell. The quasi-Z-source network is composed of $L_1$, $D_2$, $L_2$, $C_1$, $C_2$. The switch-capacitor network is composed of $D_3$, $D_4$, $D_5$, $C_3$, $C_4$ and $C_5$. $R$ is the load resistance and $u_o$ is the output voltage of the converter. There is only one power switch $Q$ in the topology.

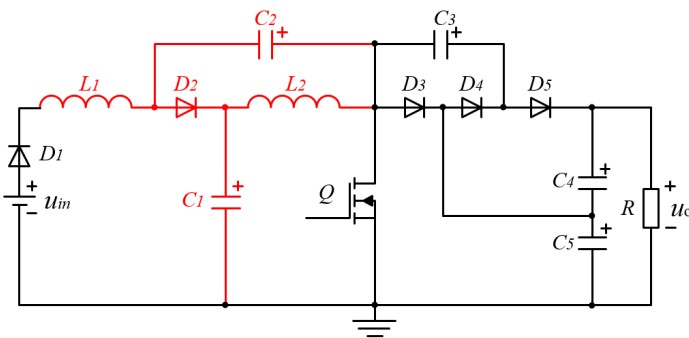

**Figure 1.** The DC–DC converter topology based on the quasi-Z-source network.

### 2.1. Topology Operation Principle Analysis

According to switching states of the power switch $Q$, the topology has two operation states. It is assumed that the inductors, capacitors and power semiconductors in the topology are ideal components. The Kirchhoff voltage law and current law are applied to analyze the topology operation principle. When the power switch $Q$ is turned on, the operation state of the circuit is as shown in Figure 2. Diodes $D_2$, $D_3$ and $D_5$ are turned off and four current loops appear in the circuit. The output voltage of fuel cell $u_{in}$ and capacitor $C_2$ charge inductor $L_1$ through diode $D_1$ and power switch $Q$. $C_1$ charges $L_2$ through $Q$. $C_5$ charges $C_3$ through $D_4$ and $Q$. $C_4$ and $C_5$ in series transfers the energy to load $R$. Equations (1) and (2) can be obtained from Figure 2.

$$\begin{cases} u_{L1on} = u_{in} + u_{C2} \\ u_{L2on} = u_{C1} \\ u_{C3} = u_{C5} \\ u_O = u_{C4} + u_{C5} \end{cases} \tag{1}$$

where $u_{C1}$, $u_{C2}$, $u_{C3}$, $u_{C4}$ and $u_{C5}$ are the capacitor voltages across $C_1$, $C_2$, $C_3$, $C_4$ and $C_5$, respectively. $u_{L1on}$ and $u_{L2on}$ are the inductor voltages across $L_1$ and $L_2$, respectively, when power switch $Q$ is turned on.

$$\begin{cases} I_{C1on} = -I_{L2} \\ I_{C2on} = -I_{L1} \\ I_{C4on} = -I_O \\ I_{C5on} = I_{C4on} - I_{C3on} \end{cases} \tag{2}$$

where $I_{L1}$ and $I_{L2}$ are, respectively, the average inductor currents of $L_1$ and $L_2$. $I_{C1on}$, $I_{C2on}$, $I_{C3on}$, $I_{C4on}$ and $I_{C5on}$ are, respectively, the average capacitor currents of $C_1$, $C_2$, $C_3$, $C_4$ and $C_5$ when the power switch $Q$ is turned on. $I_O$ is the output average current of the converter.

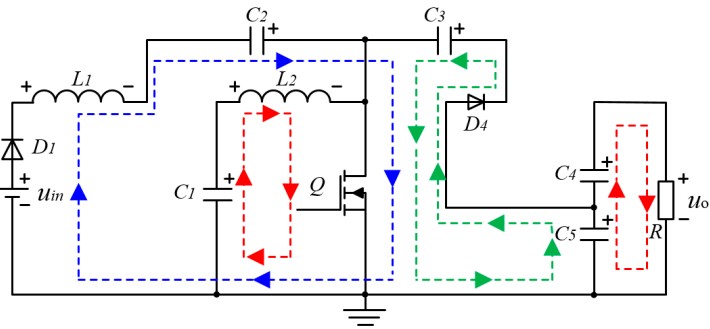

**Figure 2.** The circuit operation state when power switch $Q$ is turned on.

When the power switch $Q$ is turned off, the diode $D_4$ is turned off and four current loops also appear in the circuit. $u_{in}$ and inductor $L_1$ charge capacitor $C_1$ through diodes $D_1$ and $D_2$. $u_{in}$, $L_1$ and $L_2$ charge $C_5$ through $D_1$, $D_2$ and $D3$. $u_{in}$, $L_1$, $L_2$ and $C_3$ charge the load $R$ and the series part of $C_4$ and $C_5$ through $D_1$, $D_2$ and $D_5$. $L_2$ charges $C_2$ through $D_2$. Equations (3) and (4) can be obtained from Figure 3.

$$\begin{cases} u_{L1off} = u_{in} - u_{C1} \\ u_{L1off} = u_{in} - u_O + u_{C2} + u_{C3} \\ u_{L2off} = -u_{C2} \\ u_{C3} = u_{C4} \end{cases} \tag{3}$$

where $u_{L1\text{off}}$ and $u_{L2\text{off}}$ are, respectively, the inductor voltages across $L_1$ and $L_2$ when the power switch $Q$ is turned off.

$$\begin{cases} I_{C1\text{off}} = I_{C2\text{off}} + I_{L1} - I_{L2} \\ I_{C3\text{off}} = -I_{C4\text{off}} - I_O \\ I_{C5\text{off}} = I_{L1} - I_O - I_{C1\text{off}} \end{cases} \tag{4}$$

where $I_{C1\text{off}}$, $I_{C2\text{off}}$, $I_{C3\text{off}}$, $I_{C4\text{off}}$ and $I_{C5\text{off}}$ are, respectively, the average capacitor currents of $C_1$, $C_2$, $C_3$, $C_4$ and $C_5$ when the power switch $Q$ is turned off.

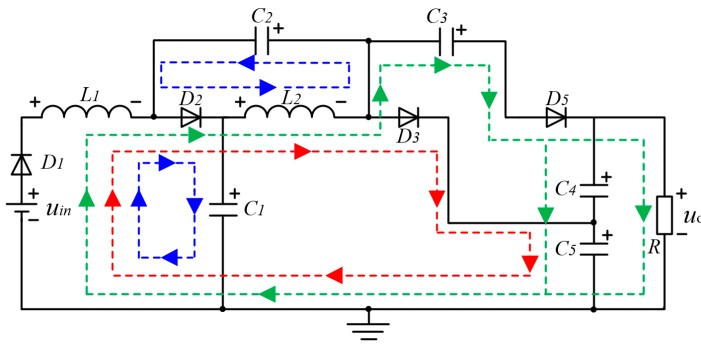

**Figure 3.** The circuit operation state when power switch $Q$ is turned off.

Equation (5) is obtained by applying the voltage-second balance principle to inductors $L_1$ and $L_2$ and Equation (6) is obtained by applying the ampere-second balance principle to capacitors $C_1$, $C_2$, $C_3$, $C_4$ and $C_5$.

$$\begin{cases} u_{L1\text{on}} \times dT + u_{L1\text{off}} \times (1-d)T = 0 \\ u_{L2\text{on}} \times dT + u_{L2\text{off}} \times (1-d)T = 0 \end{cases} \tag{5}$$

$$\begin{cases} I_{C1\text{on}} \times dT + I_{C1\text{off}} \times (1-d)T = 0 \\ I_{C2\text{on}} \times dT + I_{C2\text{off}} \times (1-d)T = 0 \\ I_{C3\text{on}} \times dT + I_{C3\text{off}} \times (1-d)T = 0 \\ I_{C4\text{on}} \times dT + I_{C4\text{off}} \times (1-d)T = 0 \\ I_{C5\text{on}} \times dT + I_{C5\text{off}} \times (1-d)T = 0 \end{cases} \tag{6}$$

where $d$ is the duty cycle. $T$ is the switch period of power switch $Q$.

According to Equations (1), (3) and (5), the voltage gain $M$ is obtained as shown in Equation (7).

$$M = \frac{u_O}{u_{\text{in}}} = \frac{2}{1-2d} \tag{7}$$

where $0 < d < 0.5$.

Assuming that the input power is equal to the output power, Equation (8) can be obtained.

$$u_{\text{in}} \times I_{\text{in}} = u_O \times I_O \tag{8}$$

where $I_{\text{in}}$ is the input average current of the converter.

According to Equations (2), (4), (6) and (8), the current relationship can be obtained as shown in Equation (9).

$$I_{L1} = I_{L2} = I_{\text{in}} = \frac{2}{1-2d} I_O \tag{9}$$

According to Figures 2 and 3 and Equations (1), (3) and (5), the voltage stresses across diodes, the power switch and capacitors are obtained as shown in Equation (10). The voltage stress across diode $D_1$ is not considered, and is always turned on.

$$\begin{cases} u_{D2} = u_{C1} + u_{C2} = u_O/2 \\ u_{D3} = u_{C5} = u_O/2 \\ u_{D4} = u_{C3} = u_O/2 \\ u_{D5} = u_{C4} = u_O/2 \\ u_Q = u_{C5} = u_O/2 \end{cases} \tag{10}$$

According to Equation (7), the converter has high gain and wide voltage input range. The duty cycle is always less than 0.5. From Equation (10), the voltage stresses across components are low, which is suitable for the DC–DC converter for fuel cell vehicle.

## 2.2. Dynamic Modeling

In this paper, the converter operates in continuous conduction mode (CCM). The state space averaging method is used to model the converter. It is assumed that the inductors, capacitors and power semiconductors are ideal components. When the power switch $Q$ is turned on, capacitor $C_3$ and capacitor $C_5$ are connected in parallel as shown in Figure 2. Equation (11) can be obtained in the loop circuit of $C_3$ and $C_5$ in Figure 2.

$$C_3 \frac{du_{C3}}{dt} = -C_5 \frac{du_{C5}}{dt} \tag{11}$$

Therefore, there is an invalid state space variable between $C_3$ and $C_5$ in the state space equation. The coupling relationship between $C_3$ and $C_5$ can be eliminated by introducing series equivalent resistance $r_1$ into the loop circuit of $C_3$ and $C_5$. Equation (11) can be written as Equation (12).

$$C_3 \frac{du_{C3}}{dt} = -\frac{u_{C5} - u_{C3}}{r_1} \tag{12}$$

where, $r_1$ is the series equivalent resistance in the loop circuit of $C_3$ and $C_5$. $r_1 = 0.1\ \Omega$.

When the power switch $Q$ is turned off, the series equivalent resistance $r_2$ ($r_2 = 0.1\ \Omega$) can be adopted to eliminate the coupling relationship between $C_1$, $C_2$, $C_3$, $C_4$ and $C_5$.

The state variables are $x = [i_{L1}, i_{L2}, u_{C1}, u_{C2}, u_{C3}, u_{C4}, u_{C5}]$. The input variable is $u = u_{in}$ and the output variable is $y = u_o$. The open-time of the power switch is $dT$ and the close-time is $(1-d)T$ in a switching period. Therefore, when power switch $Q$ is turned on, the state space equation can be obtained as Equation (13). When power switch $Q$ is turned off, the state space equation can be obtained as Equation (14).

$$\begin{cases} \dot{x} = A_{1x} + B_{1u} \\ y = C_{1x} + D_{1u} \end{cases} \tag{13}$$

$$\begin{cases} \dot{x} = A_{2x} + B_{2u} \\ y = C_{2x} + D_{2u} \end{cases} \tag{14}$$

where the coefficient matrixes $A_1$ and $A_2$ can be obtained as in Equation (15). The coefficient matrixes $B_1$, $C_1$, $D_1$, $B_2$, $C_2$ and $D_2$ can be obtained as in Equation (16).

$$A_1 = \begin{bmatrix} 0 & 0 & 0 & \frac{1}{L_1} & 0 & 0 & 0 \\ 0 & 0 & \frac{1}{L_2} & 0 & 0 & 0 & 0 \\ 0 & \frac{-1}{C_1} & 0 & 0 & 0 & 0 & 0 \\ \frac{-1}{C_2} & 0 & 0 & 0 & 0 & 0 & 0 \\ 0 & 0 & 0 & 0 & \frac{-1}{C_3 r_1} & 0 & \frac{1}{C_3 r_1} \\ 0 & 0 & 0 & 0 & 0 & \frac{-1}{C_4 R} & \frac{-1}{C_4 R} \\ 0 & 0 & 0 & 0 & \frac{1}{C_5 r_1} & \frac{-1}{C_5 r_1} & \frac{-R-r_1}{C_5 R r_1} \end{bmatrix}$$

$$A_2 = \begin{bmatrix} 0 & 0 & \frac{-1}{L_1} & 0 & 0 & 0 & 0 \\ 0 & 0 & 0 & \frac{-1}{L_2} & 0 & 0 & 0 \\ \frac{1}{C_1} & 0 & \frac{-r_1-r_2}{C_1 r_1 r_2} & \frac{-r_1-r_2}{C_1 r_1 r_2} & \frac{-1}{C_1 r_2} & \frac{1}{C_1 r_2} & \frac{r_1+r_2}{C_1 r_1 r_2} \\ 0 & \frac{1}{\bar{u}_o} & \frac{-r_1-r_2}{\bar{u}_o r_1 r_2} & \frac{-r_1-r_2}{\bar{u}_o r_1 r_2} & \frac{-1}{\bar{u}_o r_2} & \frac{-1}{C_2 r_2} & \frac{r_1+r_2}{C_2 r_1 r_2} \\ 0 & 0 & \frac{-1}{C_3 r_2} & \frac{-1}{r_2 r_2} & \frac{-1}{r_2 r_2} & \frac{1}{r_2 r_2} & \frac{1}{r_2 r_2} \\ 0 & 0 & \frac{1}{C_4 r_2} & \frac{1}{C_4 r_2} & \frac{1}{C_4 r_2} & \frac{-R-r_2}{C_4 R r_2} & \frac{-R-r_2}{C_4 R r_2} \\ 0 & 0 & \frac{r_1+r_2}{C_5 r_1 r_2} & \frac{r_1+r_2}{C_5 r_1 r_2} & \frac{1}{C_5 r_2} & \frac{-R-r_2}{C_5 R r_2} & \frac{-R r_1-r_1 r_2-R r_2}{C_5 R r_1 r_2} \end{bmatrix}$$

(15)

$$B1 = B2 = [1/L1, 0, 0, 0, 0, 0, 0]^T \quad C1 = C2 = [0\,0\,0\,0\,0\,1\,1] \quad D1 = D2 = [0] \tag{16}$$

According to Equations (13) and (14), the state space average equation is obtained as in Equation (17).

$$\begin{cases} \dot{\bar{x}} = (A_1 \bar{d} + A_2(1-\bar{d}))\bar{x} + (B_1 \bar{d} + B_2(1-\bar{d}))\bar{u} \\ \bar{y} = (C_1 \bar{d} + C_2(1-\bar{d}))\bar{x} + (D_1 \bar{d} + D_2(1-\bar{d}))\bar{u} \end{cases} \tag{17}$$

where the average values of the state variables are $\bar{x} = \left[\bar{i}_{L1}, \bar{i}_{L2}, \bar{u}_{C1}, \bar{u}_{C2}, \bar{u}_{C3}, \bar{u}_{C4}, \bar{u}_{C5}\right]$ in a switching period. $\bar{d}$ is the average duty cycle in a switching cycle. The average value of the input variable is $\bar{u} = \bar{u}_{in}$. The average value of the output variable is $\bar{y} = \bar{u}_o$.

In the small-signal modeling method, each average variable is composed of a DC component and an AC small-signal component, as shown in Equation (18).

$$\begin{cases} \bar{i}_{L1} = I_{L1} + \widetilde{i}_{L1} \\ \bar{i}_{L2} = I_{L2} + \widetilde{i}_{L2} \\ \bar{u}_{C1} = U_{C1} + \widetilde{u}_{C1} \\ \bar{u}_{C2} = U_{C2} + \widetilde{u}_{C2} \\ \bar{u}_{C3} = U_{C3} + \widetilde{u}_{C3} \\ \bar{u}_{C4} = U_{C4} + \widetilde{u}_{C4} \\ \bar{u}_{C5} = U_{C5} + \widetilde{u}_{C5} \\ \bar{u}_{in} = U_{in} + \widetilde{u}_{in} \\ \bar{u}_O = U_O + \widetilde{u}_O \\ \bar{d} = D + \widetilde{d} \end{cases} \tag{18}$$

where $I_{L1}$, $I_{L2}$, $U_{C1}$, $U_{C2}$, $U_{C3}$, $U_{C4}$, $U_{C5}$, $U_{in}$, $U_O$ and $D$ are DC steady-state components. $\widetilde{u}_{L1}, \widetilde{u}_{L2}, \widetilde{u}_{C1}, \widetilde{u}_{C2}, \widetilde{u}_{C3}, \widetilde{u}_{C4}, \widetilde{u}_{C5}, \widetilde{u}_{in}, \widetilde{u}_O$ and $\widetilde{d}$ are small-signal components.

According to Equations (17) and (18) and ignoring the second order infinitesimal terms, the small-signal model of the converter can be written as Equation (19).

$$\begin{cases} \dot{\widetilde{x}} = \widetilde{A}\widetilde{x} + \widetilde{B}_1 \widetilde{u} + \widetilde{B}_2 \widetilde{d} \\ \widetilde{y} = \widetilde{C}\widetilde{x} + \widetilde{D}\widetilde{u} \end{cases} \tag{19}$$

where the coefficient matrixes $\widetilde{A}$, $\widetilde{B}_1$, $\widetilde{B}_2$, $\widetilde{C}$ and $\widetilde{D}$ can be obtained as in Equation (20).

$$
\widetilde{A} = \begin{bmatrix}
0 & 0 & \frac{D-1}{L_1} & \frac{D}{L_1} & 0 & 0 & 0 \\
0 & 0 & \frac{D}{L_2} & \frac{D-1}{L_2} & 0 & 0 & 0 \\
\frac{1-D}{C_1} & -\frac{D}{C_1} & \frac{(r_1+r_2)(D-1)}{C_1 r_1 r_2} & \frac{(r_1+r_2)(D-1)}{C_1 r_1 r_2} & \frac{D-1}{C_1 r_2} & \frac{1-D}{C_1 r_2} & \frac{(r_1+r_2)(1-D)}{C_1 r_1 r_2} \\
-\frac{D}{C_2} & \frac{1-D}{C_2} & \frac{(r_1+r_2)(D-1)}{C_2 r_1 r_2} & \frac{(r_1+r_2)(D-1)}{C_2 r_1 r_2} & \frac{D-1}{C_2 r_2} & \frac{1-D}{C_2 r_2} & \frac{(r_1+r_2)(1-D)}{C_2 r_1 r_2} \\
0 & 0 & \frac{D-1}{r_2 r_2} & \frac{D-1}{r_2 r_2} & \frac{D(r_1-r_2)-r_1}{r_2 r_1 r_2} & \frac{1-D}{r_2 r_2} & \frac{D(r_2-r_1)+r_1}{r_2 r_1 r_2} \\
0 & 0 & \frac{1-D}{C_4 r_2} & \frac{1-D}{C_4 r_2} & \frac{1-D}{C_4 r_2} & \frac{DR-R-r_2}{C_4 R r_2} & \frac{DR-R-r_2}{C_4 R r_2} \\
0 & 0 & \frac{(r_1+r_2)(1-D)}{C_5 r_1 r_2} & \frac{(r_1+r_2)(1-D)}{C_5 r_1 r_2} & \frac{D(r_2-r_1)+r_1}{C_5 r_1 r_2} & \frac{DR-R-r_2}{C_5 R r_2} & \frac{DRr_1-Rr_1-rr_1-Rr_2}{C_5 R r_1 r_2}
\end{bmatrix}
\quad
\widetilde{B}_1 = \begin{bmatrix} \frac{1}{L_1} \\ 0 \\ 0 \\ 0 \\ 0 \\ 0 \\ 0 \end{bmatrix}
$$

$$
\widetilde{B}_2 = \begin{bmatrix}
0 & 0 & \frac{1}{L_1} & \frac{1}{L_1} & 0 & 0 & 0 \\
0 & 0 & \frac{1}{L_2} & \frac{1}{L_2} & 0 & 0 & 0 \\
-\frac{1}{C_1} & -\frac{1}{C_1} & \frac{r_1+r_2}{C_1 r_1 r_2} & \frac{r_1+r_2}{C_1 r_1 r_2} & \frac{1}{C_1 r_2} & -\frac{1}{C_1 r_2} & -\frac{r_1+r_2}{C_1 r_1 r_2} \\
-\frac{1}{C_2} & -\frac{1}{C_2} & \frac{r_1+r_2}{C_2 r_1 r_2} & \frac{r_1+r_2}{C_2 r_1 r_2} & \frac{1}{C_2 r_2} & -\frac{1}{C_2 r_2} & -\frac{r_1+r_2}{C_2 r_1 r_2} \\
0 & 0 & \frac{1}{r_2 r_2} & \frac{1}{r_2 r_2} & \frac{r_1-r_2}{r_2 r_1 r_2} & -\frac{1}{r_2 r_2} & \frac{r_2-r_1}{r_2 r_1 r_2} \\
0 & 0 & -\frac{1}{C_4 r_2} & -\frac{1}{C_4 r_2} & \frac{1}{C_4 r_2} & \frac{R}{C_4 R r_2} & \frac{R}{C_4 R r_2} \\
0 & 0 & -\frac{r_2-r_1}{C_5 r_1 r_2} & -\frac{r_1+r_2}{C_5 r_1 r_2} & \frac{r_2-r_1}{C_5 r_1 r_2} & \frac{R}{C_5 R r_2} & \frac{Rr_1}{C_5 R r_1 r_2}
\end{bmatrix}
\begin{bmatrix} I_{L1} \\ I_{L2} \\ U_{C1} \\ U_{C2} \\ U_{C3} \\ U_{C4} \\ U_{C5} \end{bmatrix}
\quad \widetilde{D} = [0] \quad \widetilde{C} = [0000011]
$$

(20)

## 3. Component Parameters and Controller Design

### 3.1. Component Parameters Design

The converter parameters are shown in Table 1. According to Equation (9), the inductor currents of $L_1$ and $L_2$ are consistent. The inductor $L_1$ is taken as an example for calculation. The calculation Equation of inductor $L_1$ can be obtained as shown in Equation (21).

$$
L_1 = U_{L1} \frac{dt}{di_{L1}}
\tag{21}
$$

where $di_{L1} = \Delta IL_1$, $u_{L1} = u_{\text{in}} + u_{C2}$, $dt = d \times T = d/f$ when the power switch $Q$ is turned on. $f$ is the switching frequency.

**Table 1.** Converter parameters table.

| Parameters | Values | Units |
|---|---|---|
| Rated Power $P$ | 400 | W |
| Input voltage $u_{\text{in}}$ | 40~120 | V |
| Output voltage $u_O$ | 400 | V |
| Load resistance $R$ | 400 | $\Omega$ |
| Switching frequency $f$ | 20 | kHz |

According to Equations (1), (3), (5) and (21), the inductors $L_1$ and $L_2$ can be obtained as in Equation (22).

$$
L_1 = L_2 = \frac{d(1-d)u_{\text{in}}}{(1-2d)rI_{L1}f}
\tag{22}
$$

where $r$ is the ripple rate of the inductor current. $r = \Delta I_{L1}/I_{L1}$.

The calculation Equation of capacitor can be obtained as in Equation (23).

$$
C = i_C \frac{dt}{du_C}
\tag{23}
$$

where $i_C$ is the capacitor current and $u_C$ is the capacitor voltage. $du_C = \Delta u_C$. $\Delta u_C$ is the capacitor ripple voltage.

According to Equations (1)–(6) and (23), the calculation equations of capacitors are obtained as in Equation (24).

$$\begin{cases} C_1 = \frac{2dI_O}{(1-2d)\Delta_{UC1}f} \\ C_2 = \frac{2d^2I_O}{(1-2d)\Delta_{UC2}f} \\ C_3 = \frac{2I_O}{\Delta_{UC3}f} \\ C_4 = \frac{4dI_O}{(1-2d)^2\Delta_{UC4}f} \\ C_5 = \frac{(1+d)I_O}{\Delta_{UC5}f} \end{cases} \tag{24}$$

The inductor and capacitor parameters are finally obtained as shown in Table 2.

**Table 2.** Parameters table of inductors and capacitors.

| Parameters | Values | Units |
|:---:|:---:|:---:|
| Inductors $L_1$, $L_2$ | 800 | uH |
| Capacitors $C_1$, $C_2$, $C_3$, $C_4$, $C_5$ | 680 | uF |

### 3.2. Feedforward Compensation Network and Feedback Control Design

The block diagram of the closed-loop feedback control system is shown in Figure 4. $G_{uo-d}(s)$ is the transfer function from duty cycle to output voltage. $G_{PWM}(s)$ is the transfer function of the pulse width modulator and $G_{FB}(s)$ is the feedback network transfer function. In the prototype, the hall sensor is adopted to measure the output voltage. According to the variable ratio of the hall sensor, $G_{FB}(s) = 1/200$ is obtained.

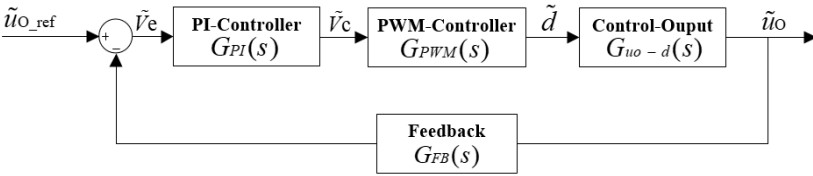

**Figure 4.** Block diagram of feedback control system.

According to Equation (19) and Tables 1 and 2, when the input voltage is 40 V, $G_{uo-d}(s)$ can be obtained as shown in Equation (25) by using the ss2tf function in Matlab.

$$G_{uo-d}(s) = \left.\frac{\tilde{u}_o(s)}{\tilde{d}(s)}\right|_{\widetilde{uin}=0} = \frac{9.41\times10^{-16}s^6+5.56\times10^{-11}s^5+6.14\times10^{-7}s^4+1.83\times10^{-3}s^3+1.15s^2+317s+37553}{1\times10^{-22}s^7+8.24\times10^{-18}s^6+1.04\times10^{-13}s^5+3.37\times10^{-10}s^4+1.95\times10^{-7}s^3+5.94\times10^{-4}s^2+6.69\times10^{-3}s+6.2} \tag{25}$$

The system feedback controller adopts the proportional plus integral controller (PI controller), as shown in Equation (26).

$$G_{PI}(s) = K_p + K_i/s \tag{26}$$

where, $K_P$ and $K_i$ are proportional and integral coefficients, respectively.

The feedback controller can eliminate the steady-state error of the output voltage and adjust the system dynamic performance. However, when input voltage disturbance occurs, the system does not adjust the duty cycle to maintain the stability of the output voltage until the output voltage fluctuation is fed back to the control end. Due to the "soft" output characteristic of the fuel cell, the feedforward compensation network should be adopted for the DC–DC converter of fuel cell vehicles. Therefore, a composite controller is proposed with input voltage feedforward compensation and feedback control. The block diagram of the system is as shown in Figure 5. $G_{uo-ui}(s)$ is the transfer function from input

to output and $G_f(s)$ is the feedforward compensation transfer function. According to Equation (19), the $G_{uo-ui}(s)$ can be obtained as in Equation (27), when the input voltage is 40 V.

$$G_{uo-ui}(s) = \left.\frac{\widetilde{u}_o(s)}{\widetilde{u}_i(s)}\right|_{\widetilde{d}=0} = \frac{-1.02\times10^{-32}s^6+-9.54\times10^{-29}s^5+9.73\times10^{-13}s^4+1.14\times10^{-8}s^3+3.55s^{-5}+0.021s+62}{1\times10^{-22}s^7+8.24\times10^{-18}s^6+1.04\times10^{-13}s^5+3.37\times10^{-10}s^4+1.95\times10^{-7}s^3+5.94\times10^{-4}s^2+6.69\times10^{-3}s+6.2} \tag{27}$$

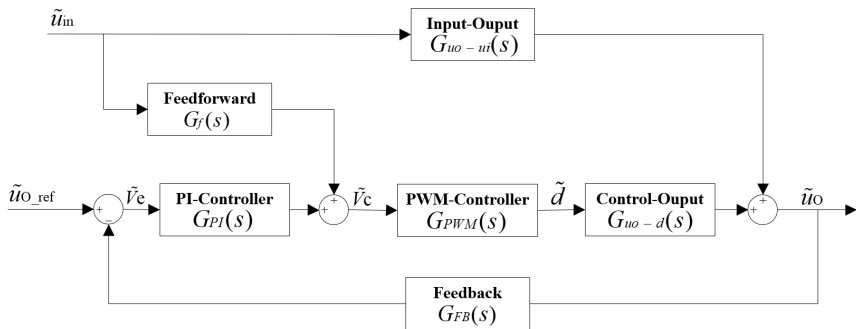

**Figure 5.** Block diagram of feedforward compensation and the feedback control system.

According to Figure 5, the feedforward compensation transfer function $G_f(s)$ is as in Equation (28).

$$G_f(s) = \frac{G_{uo-ui}(s)}{G_{PWM}(s)G_{uo-d}(s)} \tag{28}$$

where $G_{PWM}(s)$ is a proportional element and is equal to 1 in the simulation system.

According to Equations (25) and (27), the calculation result of Equation (28) is shown in Equation (29). The feedforward compensation network and feedback controller work together to adjust the duty cycle. The feedforward compensation network can reduce the output voltage fluctuation caused by input voltage disturbance, which is beneficial for improving the performance of the DC–DC converter for fuel cell vehicles.

$$G_f(s) = \frac{-1.02\times10^{-32}s^6+-9.54\times10^{-29}s^5+9.73\times10^{-13}s^4+1.14\times10^{-8}s^3+3.55s^{-5}+0.021s+62}{9.41\times10^{-16}s^6+5.56\times10^{-11}s^5+6.14\times10^{-7}s^4+1.83\times10^{-3}s^3+1.15s^2+317s+37553} \tag{29}$$

## 4. Simulation and Experimental Results

In this section, the simulation model in Matlab/Simulink and the experimental prototype are built to validate the performance of the proposed composite control strategy. The simulation and experiment parameters are as shown in Tables 1 and 2.

### 4.1. Simulation Results

The load resistance is 400 Ω and the reference output voltage is 400 V. When the input voltage jumps between 50 V and 60 V, the output voltage $u_O$ and input voltage $u_{in}$ are as shown in Figure 6.

Figure 6a is the output voltage under input voltage disturbance with only feedback controller. When input voltage disturbance occurs, the output voltage maximum excursion is around 7%. Figure 6b is the output voltage under input voltage disturbance with feedforward compensation and feedback controller. The output voltage maximum excursion is less than 1.5%. The result indicates that the feedforward compensation network improve the system's ability to resist input voltage disturbance.

In the process of fuel cell vehicle driving, the vehicle demand power changes frequently, which requires fast dynamic response performance and an excellent ability to resist load disturbances for the converter. The output voltage is as shown in Figure 7 during load 50% step change. When the input voltage is 40 V and the reference output voltage is 400 V, the load resistance jumps between 400 Ω and 200 Ω. In Figure 7, it can be observed that the output voltage maximum excursion is less than 3% and the adjustment time is less than 30 ms during load 50% step change.

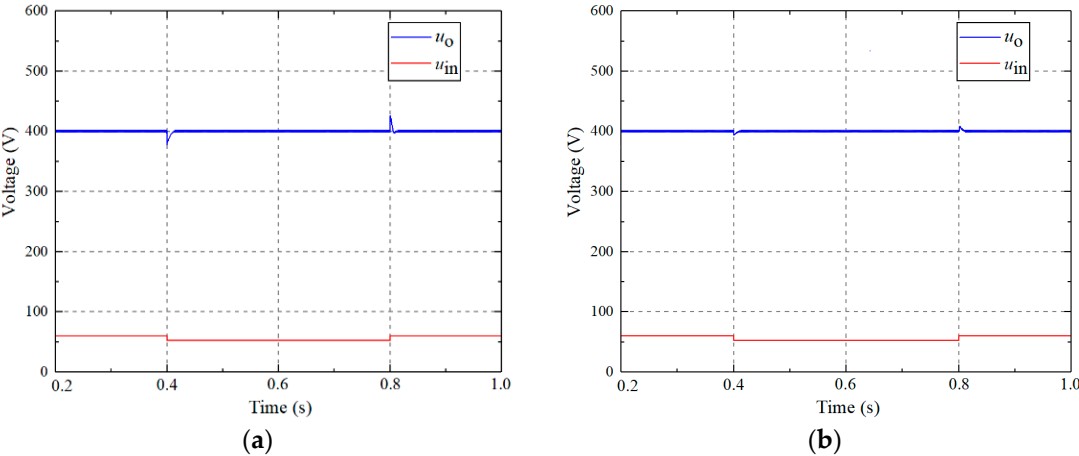

**Figure 6.** The output voltage variations under input voltage disturbance. (**a**) With only feedback controller; (**b**) with feedback controller and feedforward compensation.

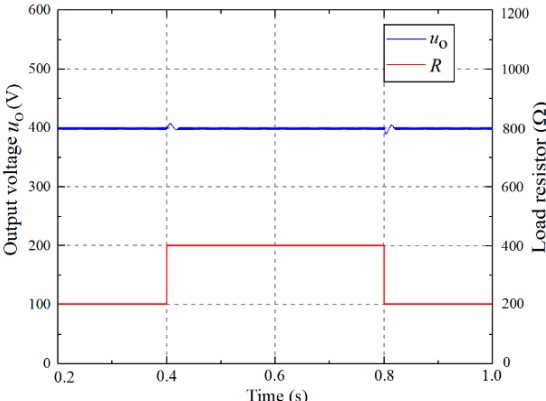

**Figure 7.** The output voltage during load 50% step change.

The worldwide harmonised light vehicle test procedure (WLTP) is adopted to test the converter. The WLTP consists of low speed, medium speed, high speed and extra-high speed, as shown in Figure 8 [26,27]. The parameters of the fuel cell used in the simulation are shown in Table 3. The relationship between the demand power and the speed of the vehicle is as shown in Equation (30).

$$P_{\text{load}} = (M\frac{dv}{dt} + MgC_{\text{r}} + \frac{\rho_{\text{air}}Sv^2}{2})v \tag{30}$$

where $v$ is the vehicle speed. $P_{\text{load}}$ is the demand power of the vehicle. The vehicle weight is $M$ = 300 kg. The rolling resistance coefficient is $Cr$ = 0.001. The front windward area of the vehicle is $S$ = 1 m². The gravity acceleration is $g$ = 9.81 m/s². The wheel radius is $Rad$ = 0.14 m. The volume density of air is $\rho_{\text{air}}$ = 1.2kg/m³ [28].

The fuel cell vehicles typically use the compound energy system, which consists of a fuel cell and an ultracapacitor. The bi-directional DC–DC converter is used between the ultracapacitor and the DC bus to provide instantaneous power or to recover braking energy. When the required power $P_{\text{load}}$ in Equation (30) is less than zero, the vehicle is recovering energy. The quasi-Z-source network DC–DC converter in this paper is used between the fuel cell and the DC bus only to provide power for the vehicle. Therefore, the vehicle load demand power $P_{\text{load}}$ is limited to zero when the $P_{\text{load}}$ is less than zero. Then, the vehicle load demand power $P_{\text{load}}$ is scaled down in order to be less than the converter's rated power. The simulation result of the converter load current is shown in Figure 9. In order to track the load current variation of the converter, the output current of the fuel cell is shown in Figure 10. The output voltage of the fuel cell is shown in Figure 11.

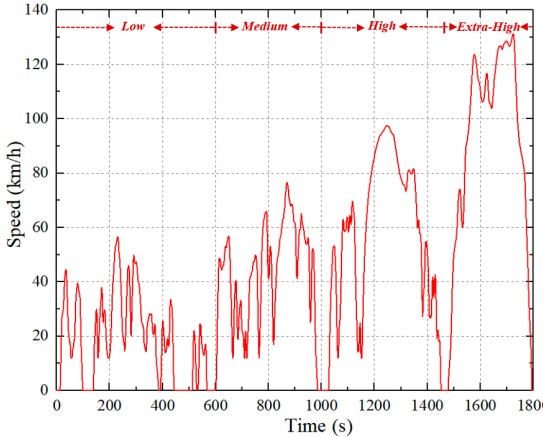

**Figure 8.** Worldwide harmonised light vehicle test procedure (WLTP).

**Table 3.** Fuel cell parameters table in simulation.

| Parameters | Values | Units |
|---|---|---|
| Output power $P_{FC}$ | 1200 | W |
| Maximum output voltage $V_{FC}$ | 58 | V |
| Maximum output current $I_L$ | 20 | A |
| Maximum temperature $T\_max$ | 373 | K |
| Initial temperature $T\_initial$ | 307 | K |
| Partial pressure of hydrogen $PH_2$ | 1.5 | atm |
| Partial pressure of oxygen $PO_2$ | 1 | atm |
| Partial pressure of water $PHO_2$ | 1 | atm |

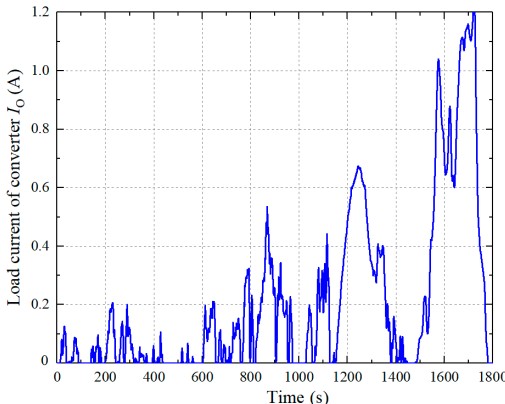

**Figure 9.** The converter load current under the worldwide harmonised light vehicle test procedure (WLTP).

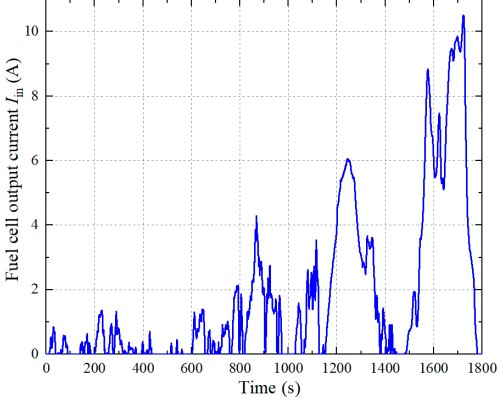

**Figure 10.** The fuel cell output current under the WLTP.

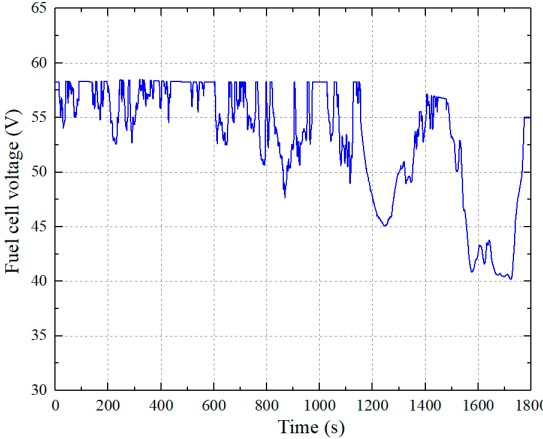

**Figure 11.** The output voltage of a fuel cell under the WLTP.

The converter output voltage is shown in Figure 12. In order to maintain a stable output voltage of the converter, when the fuel cell output voltage drops, the duty cycle needs to be raised correspondingly. The duty cycle variation curve is shown in Figure 13.

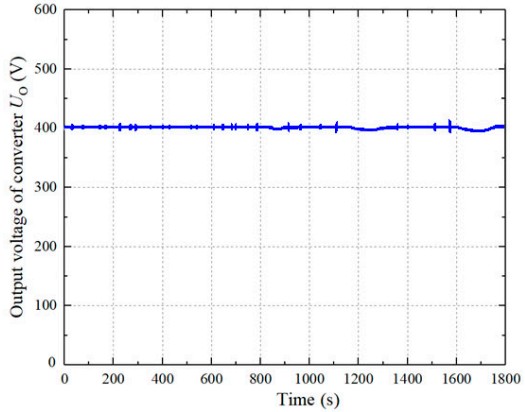

**Figure 12.** The output voltage of a converter under the WLTP.

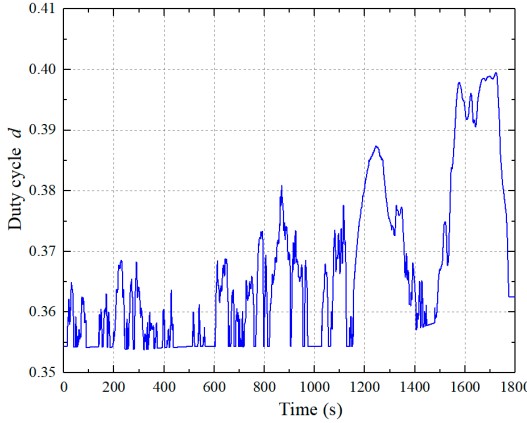

**Figure 13.** The duty cycle variation curve.

*4.2. Experimental Results*

A 400 W experiment prototype was built as shown in Figure 14. The parameters of the prototype are shown in Tables 1 and 2. The TMS320F28335 is adopted as controller and the MOSFET (IXTK102N30P)

is adopted as power switch. The experiment platform is shown in Figure 15. The input voltage and output voltage are measured by the voltage hall sensors.

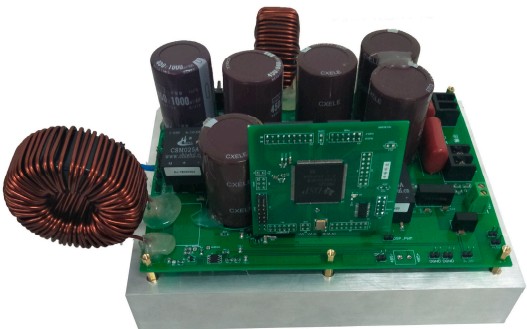

**Figure 14.** The experiment prototype.

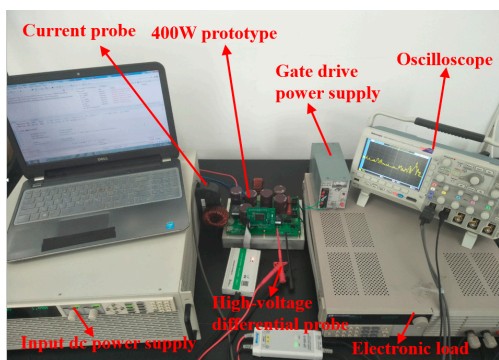

**Figure 15.** The experiment platform.

During the input voltage disturbance experiment, the output voltage reference value is set at 400 V and the input voltage jumps between 60 V and 50 V. The output voltage waveform is shown in Figure 16.

In Figure 16a, the maximum output voltage excursion is around 8% without feedforward compensation. With feedforward compensation, the maximum output voltage excursion is less than 2%, as in Figure 16c. The output voltages in Figure 16b,d are measured by using the oscilloscope AC coupling mode. From Figure 16, it can be verified that the feedforward compensation improves the system's ability to resist input voltage disturbance.

During the 50% load step disturbance experiment, the input voltage is fixed at 40 V and the output voltage reference value is set at 400 V. The load resistance is step changed between 400 Ω and 200 Ω by using the electronic load. The input current and output voltage waveforms of the converter are shown in Figure 17. The input current of the converter can quickly follow the output load variation, providing demand power for the load. The maximum output voltage excursion is less than 5% and the adjustment time is less than 30 ms. The output voltage fluctuation in Figure 17b is measured by using the oscilloscope AC coupling mode. The maximum excursion of the output voltage is slightly higher than that in the simulation; this is caused by the parasitic parameters of the circuits and components. The system shows good dynamic performance during load disturbance.

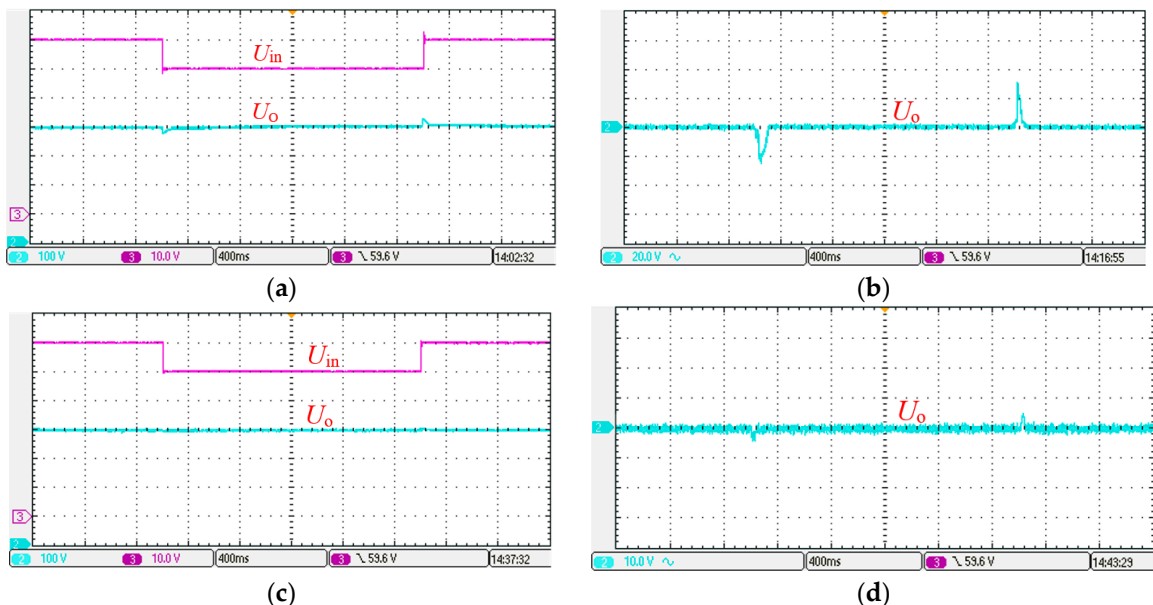

**Figure 16.** The output voltage waveform during input voltage disturbance. (**a**) Without feedforward compensation (DC coupling measurement); (**b**) without feedforward compensation (AC coupling measurement); (**c**) with feedforward compensation (DC coupling measurement); (**d**) with feedforward compensation (AC coupling measurement).

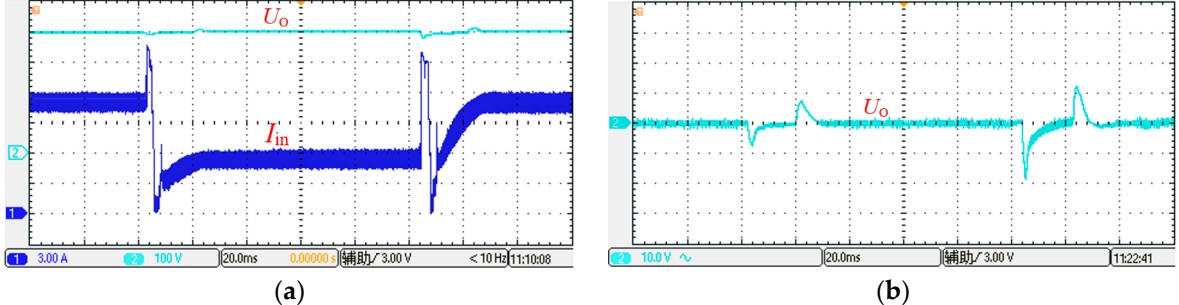

**Figure 17.** The output voltage and input current waveforms of converter during 50% load disturbance experiment. (**a**) The converter output voltage (oscilloscope DC coupling mode) and input current; (**b**) the converter output voltage (oscilloscope AC coupling mode).

The converter is tested under the WLTP. In Figure 15, a programmable power supply is adopted to simulate the variation rule of fuel cell output voltage under the WLTP according to the simulation results in Figure 11. According to Equation (30), the demand power of fuel cell vehicles under the WLTP is programmed in the electronic load. From Figure 8, the total time of the WLTP driving cycle is 1750 s. The first 1000 s is the low speed and medium speed driving cycles. Compared with the high speed and extra-high speed driving cycles in the last 750 s, the power demand in the low speed and medium speed driving cycles is smaller. Therefore, the low speed and medium speed driving cycles of the WLTP are simulated in the electronic load and the total test time is scaled down to 200 s as shown in Figure 18.

According to the fluctuation rule of fuel cell output voltage under the WLTP driving cycles in Figure 11, the fuel cell output voltage waveform is obtained by using a programmable power supply, as shown in Figure 19. The overall decreasing trend of fuel cell output voltage is simulated in the last 80 s in Figure 19.

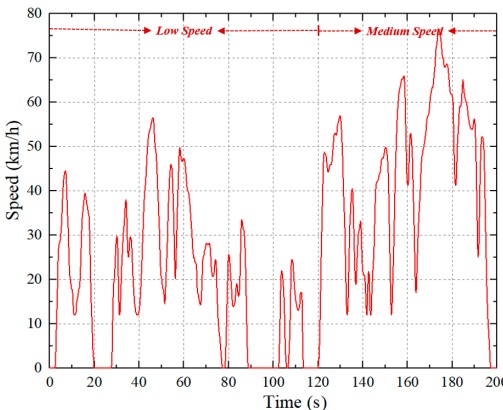

**Figure 18.** The low speed and medium speed driving cycles used in the experiment.

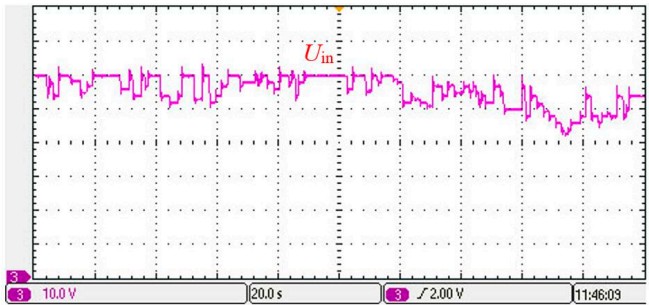

**Figure 19.** The fuel cell output voltage under the WLTP obtained by using a programmable power supply.

The load current waveform is shown in Figure 20 and changes with the load demand power. The output current waveform of the fuel cell is shown in Figure 21 and can follow the load current variation. Due to the "soft" output characteristic of the fuel cell, the fuel cell output voltage in the last 80 s reduces overall as shown in Figure 19. In order to maintain the stability of the converter output demand power, the output current of the fuel cell increases accordingly in the last 80 s, as shown in Figure 21.

Due to the proposed composite controller, the converter output voltage was maintained at 400 V during the WLTP, as shown in Figure 22. With the fuel cell input voltage fluctuation in Figure 19, in order to maintain the stability of the output voltage, the duty cycle from TMS320F28335 needs to be changed all the time, as shown in Figure 23.

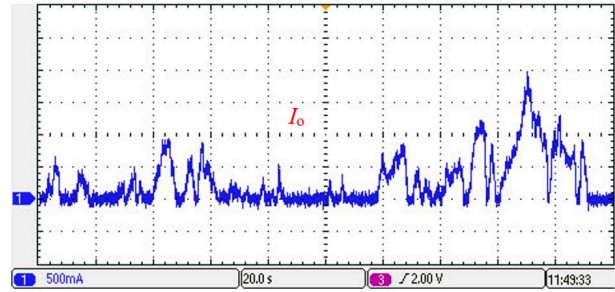

**Figure 20.** The converter load current waveform under the WLTP.

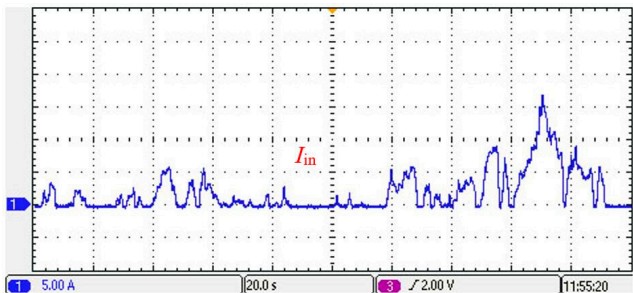

**Figure 21.** The output current waveform of the fuel cell under the WLTP.

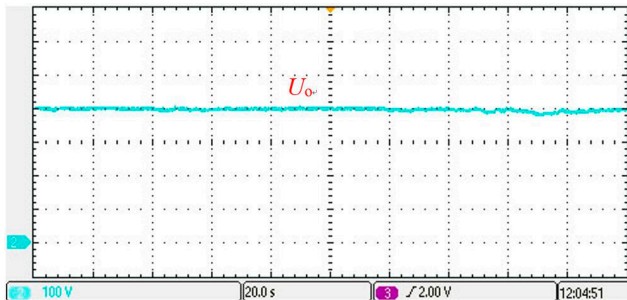

**Figure 22.** The converter output voltage waveform under the WLTP.

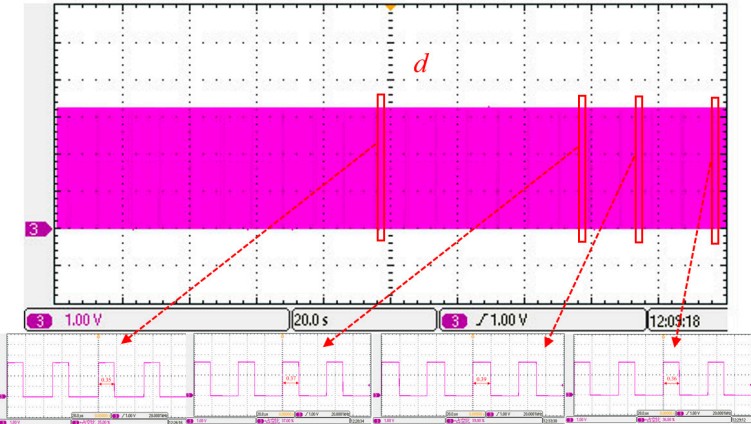

**Figure 23.** The duty cycle variation from TMS320F28335 under the WLTP.

## 5. Conclusions

Considering the power conversion requirements in the power system of fuel cell vehicles, the quasi-Z-source network DC–DC converter is adopted. The theoretical voltage gain can reach $2/(1-2d)$. When the duty cycle varies from 0.4 to 0.45, the converter voltage gain varies from 10 to 20. Compared with other DC–DC converters, with this converter topology it is easier to achieve high gain, low component stresses and wide voltage input range. When high gain is achieved, the duty cycle is always less than 0.5 and the extreme duty cycle does not occur.

Considering the "soft" output characteristic of the fuel cell, a composite controller including feedforward compensation and feedback control is designed for the quasi-Z-source network DC–DC converter. In the simulation and the experiment, the output voltage maximum excursion was less than 2% under the input voltage disturbance. The output voltage maximum excursion is less than 5% and the adjustment time is less than 30 ms under load 50% step change. Simulation and experiment results show that the proposed composite control strategy shows good dynamic response performance during load disturbance and has a good ability to resist input voltage disturbance.

Moreover, the converter was tested under the WLTP driving cycles. The results show that the converter can quickly track the load demand power variation and maintain the output voltage stability under the WLTP driving cycles. The effectiveness of the converter's actual operation performance has been verified. The composite control strategy improves the system's ability to resist voltage disturbance and dynamic response performance, which is useful for the fuel cell vehicle DC–DC converter system.

**Author Contributions:** M.Z. and X.W. conceived the control method and revised the manuscript. M.Y. performed the simulation and the experiment. J.F. wrote the full manuscript.

**Funding:** This research was funded by the National Key Technologies R&D Program of MOST (2018YFB0105403).

**Conflicts of Interest:** The authors declare no conflict of interest.

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
