# Peer review of "Research on Composite Control Strategy of Quasi-Z-Source DC–DC Converter for Fuel Cell Vehicles"

_applsci, doi:10.3390/app9163309_

Round 1
Reviewer 1 Report
Very intersting results about the output voltage stability under NEDC driving cycles. This type of converter is very usefull when the power source is a fuel cell. My recomendation to the autors is repit the simulation with the new cycle WLTP and compare the results.
Reviewer 2 Report
The manuscript entitled “Research on composite control strategy of quasi-Z source DC-DC converter for fuel cell vehicles” aims to describe the DC-DC converter for fuel cell vehicles requires high-gain and wide voltage input range to boost the voltage of the fuel cell based on a quasi-Z-source network DC-DC converter.
The research show that the proposed composite controller effectively enhances the converter's ability to resist input and load disturbance, and improves the dynamic response performance of DC-DC converter for fuel cell vehicles.
The research paper is interesting and the research is well designed but the converter is tested under NEDC cycle. The research should refer to the WLTP procedure. The United Nations Economic Commission for Europe (UNECE) to replace the new European driving cycle (NEDC) as the European vehicle homologation procedure developed this new protocol. Its final version was released in 2015. One of the main goals of the WLTP is to better match the laboratory an estimate of fuel consumption and emissions with the measures of an on-road driving condition.
From January 1, 2019 all cars in car dealerships should meet WLTP standards.
For this reason, I suggest:
- to refer to the WLTP standard, instead to the NEDC standard
- to add reference to the WLTP standards
Round 2
Reviewer 2 Report
The authors have taken into account the reviewer comments.
Moreover, they added the literature items.
